# Extractive Summarization via ChatGPT for Faithful Summary Generation

**Haopeng Zhang**    **Xiao Liu**    **Jiawei Zhang**

IFM Lab, Department of Computer Science, University of California, Davis, CA, USA
`haopeng,xiao,jiawei@ifmlab.org`

## Abstract

Extractive summarization is a crucial task in natural language processing that aims to condense long documents into shorter versions by directly extracting sentences. The recent introduction of large language models has attracted significant interest in the NLP community due to its remarkable performance on a wide range of downstream tasks. This paper first presents a thorough evaluation of ChatGPT's performance on extractive summarization and compares it with traditional fine-tuning methods on various benchmark datasets. Our experimental analysis reveals that ChatGPT exhibits inferior extractive summarization performance in terms of ROUGE scores compared to existing supervised systems, while achieving higher performance based on LLM-based evaluation metrics. In addition, we explore the effectiveness of in-context learning and chain-of-thought reasoning for enhancing its performance. Furthermore, we find that applying an extract-then-generate pipeline with ChatGPT yields significant performance improvements over abstractive baselines in terms of summary faithfulness. These observations highlight potential directions for enhancing ChatGPT's capabilities in faithful summarization using two-stage approaches.

## 1 Introduction

Document summarization aims to compress text material while retaining its most salient information. With the increasing amount of publicly available text data, automatic summarization approaches have become increasingly important. These approaches can be broadly classified into two categories: abstractive and extractive summarization. While abstractive methods (Nallapati et al., 2016; Gupta and Gupta, 2019) have the advantage of producing flexible and less redundant summaries, they often struggle with generating ungrammatical or even nonfactual contents (Kryściński et al., 2019; Zhang et al., 2022b). In contrast, extractive summarization directly selects sentences from the source document to form the summary, resulting in summaries that are grammatically correct and faithful to the original text.

The growing interest in applying advanced large language models (LLM) such as ChatGPT [1] for text summarization tasks has sparked significant attention. A recent study by (Goyal et al., 2022) compared GPT-3 with traditional fine-tuning methods and found that, despite lower Rouge scores, human annotators preferred the GPT-3 generated text. Another study by (Zhang et al., 2023d) conducted a comprehensive analysis of large language models for news summarization and found that the generated summaries were comparable to those produced by humans. However, existing research (Yang et al., 2023; Luo et al., 2023) has only focused on abstractive summary approaches, and the performance of ChatGPT for extractive summarization remains an open question. Moreover, the hallucination problem has dramatically hindered the practical use of abstractive summarization systems, highlighting the need to explore extractive summarization with LLMs for faithful summaries.

In this study, we comprehensively evaluate ChatGPT's performance on extractive summarization and investigate the effectiveness of in-context learning and chain-of-thought explanation approaches. Our experimental analysis demonstrates that ChatGPT exhibits inferior extractive summarization performance in terms of ROUGE scores compared to existing supervised systems, while achieving higher performance based on LLM-based evaluation metrics. Additionally, we observe that using an extract-then-generate pipeline with ChatGPT yields large performance improvements over abstractive baselines in terms of summary faithfulness.

The main contributions of this paper are: **1)** This study represents the first attempt to extend the ap-

---

[1] https://chat.openai.com/chat

plication of ChatGPT to extractive summarization and evaluate its performance. **2)** We investigate the effectiveness of in-context learning and chain-of-thought reasoning approaches for extractive summarization using ChatGPT. **3)** We further extend the extraction step to abstractive summarization and find that the extract-then-generate framework could improve the generated summary faithfulness by a large margin compared to abstractive-only baselines without hurting summary qualities.

## 2  Related Work

Most extractive summarization works formulate the task as a sequence classification problem and use sequential neural models with diverse encoders such as recurrent neural networks (Cheng and Lapata, 2016; Nallapati et al., 2016) and pre-trained language models (Liu and Lapata, 2019; Zhang et al., 2023b). Another group of works formulated extractive summarization as a node classification problem and applied graph neural networks to model inter-sentence dependencies (Xu et al., 2019; Wang et al., 2020; Zhang et al., 2022a, 2023a).

Several studies also explored the use of large language models (Brown et al., 2020) for summarization. Goyal et al. (2022) found that while the former obtained slightly lower Rouge scores, human evaluators preferred them. Likewise, Zhang et al. (2023d) reported that large language model-generated summaries were on par with human-written summaries in the news domain. In addition, Yang et al. (2023) explored the limits of ChatGPT on query-based summarization other than generic summarization. Luo et al. (2023) explored the use of ChatGPT as a factual inconsistency evaluator for abstractive text summarization. Zhang et al. (2023c) proposed a self-evaluation and revisement framework with ChatGPT. While most of the existing research has focused on abstractive summarization, this work aims to investigate the applicability of ChatGPT to extractive summarization and examine whether extractive methods could enhance abstractive summarization faithfulness.

## 3  Methods

### 3.1  Task Formulation

Extractive summarization systems form a summary by identifying and concatenating the most salient sentences from a given document. These approaches have gained widespread traction in various real-world applications owing to their ability to produce accurate and trustworthy summaries devoid of grammatical inconsistencies.

Formally, given a document $d$ consisting of $n$ sentences, the goal of an extractive summarization system is to produce a summary $s$ comprising of $m(m \ll n)$ sentences, by directly extracting relevant sentences from the source document. Most existing work formulates it as a sequence labeling problem, where the sentences are selected by model $M$ based on the probability of whether it should be included in the summary $s$:

$$\hat{s} = \arg \max_s p_M(s \mid d). \tag{1}$$

In the training of supervised summarization models, it is common to employ a greedy algorithm, as described in (Nallapati et al., 2017), to generate extractive ground-truth labels (ORACLE) by selecting multiple sentences that maximize the ROUGE score compared to the gold summary.

### 3.2  In-context Learning

Recent studies have shown that large language models have strong few-shot performance on various downstream tasks, known as in-context learning (ICL) (Brown et al., 2020). The standard ICL prompts a large language model, $M$, with a set of $k$ exemplar document-summary pairs and predicts a summary $\hat{s}$ for the document by:

$$\hat{s} = \arg \max_s p_M(s \mid d, \{(d^1, s^1)...(d^k, s^k)\}). \tag{2}$$

Besides simple input-output pairs, previous works also show that including explanations and chain-of-thought (COT) reasoning in prompts (Nye et al., 2021; Wei et al., 2022) also benefits language models, represented as:

$$\hat{s} = \arg \max_s p_M(s \mid d, C), \tag{3}$$

where $C = \{(d^1, e^1, s^1)...(d^k, e^k, s^k)\}$ is the set of input-explanation-output triplets in prompts. Besides zero-shot setting, this study also investigates the impact of in-context learning on extractive summarization, with and without explanations.

### 3.3  Extract-abstract Summarization

It is not new to use extractive summaries to guide abstractive summary generations (Dou et al., 2020; Wang et al., 2022). Here we also propose to use LLM in a two-stage manner: extract salient sentences to form extractive summaries ($s^E$) first, and

| Models | CNN/DM | | | | XSum | | | |
|---|---|---|---|---|---|---|---|---|
| | R1 | R2 | RL | G-EVAL | R1 | R2 | RL | G-EVAL |
| SOTA-Ext | **44.41** | **20.86** | **40.55** | 3.28 | **24.86** | 4.66 | **18.41** | 2.60 |
| ChatGPT-Ext | 39.25 | 17.09 | 25.64 | 3.24 | 19.85 | 2.96 | 13.29 | 2.67 |
| + context | 42.38 | 17.27 | 28.41 | **3.30** | 17.49 | 3.86 | 12.94 | 2.69 |
| + reason | 42.26 | 17.02 | 27.42 | 3.10 | 20.37 | **4.78** | 14.21 | **2.89** |
| SOTA-Abs | **47.78** | **23.55** | **44.63** | 3.25 | **49.07** | **25.13** | **40.40** | 2.79 |
| ChatGPT-Abs | 38.48 | 14.46 | 28.39 | **3.46** | 26.30 | 7.53 | 20.21 | **3.47** |
| Models | Reddit | | | | PubMed | | | |
| | R1 | R2 | RL | G-EVAL | R1 | R2 | RL | G-EVAL |
| SOTA-Ext | **25.09** | **6.17** | **20.13** | 1.82 | **41.21** | **14.91** | **36.75** | 2.03 |
| ChatGPT-Ext | 21.40 | 4.69 | 14.62 | **1.87** | 36.15 | 11.94 | 25.30 | 2.12 |
| + context | 22.32 | 4.86 | 14.63 | 1.83 | 36.78 | 11.86 | 25.19 | 2.14 |
| + reason | 21.87 | 4.52 | 14.65 | 1.83 | 37.52 | 12.78 | 26.36 | **2.18** |
| SOTA-Abs | **32.03** | **11.13** | **25.51** | 1.87 | **45.09** | **16.72** | **41.32** | **2.78** |
| ChatGPT-Abs | 24.64 | 5.86 | 18.54 | **2.43** | 36.05 | 12.11 | 28.46 | 2.70 |

Table 1: Summarization results on four benchmark datasets. '+context' and '+reason' refer to ChatGPT with three in-context examples and human reasoning. The best results in both extractive and abstractive settings are in bold.

| Dataset | Domain | Doc #words | Sum #words | #Ext |
|---|---|---|---|---|
| Reddit | Media | 482.2 | 28.0 | 2 |
| XSum | News | 430.2 | 23.3 | 2 |
| CNN/DM | News | 766.1 | 58.2 | 3 |
| PubMed | Paper | 444 | 209.5 | 6 |

Table 2: Detailed statistics of the datasets. Doc # words and Sum # words refer to the average word number in the source document and summary. # Ext refers to the number of sentences to extract.

then ask the LLM to generate summaries guided by the extractive summaries, represented as:

$$p(s \mid d) = \prod_{t=1}^{T} p\left(s_t \mid s_{<t}, d, s^E\right), \qquad (4)$$

where $s_{<t}$ denotes the previous generated tokens before step $t$. We explore the extract-then-generate pipeline in this study, aiming to alleviate the hallucination problems in LLM summary generation.

## 4 Experiments and Analysis

### 4.1 Experiment Settings

**Datasets:** We chose four publicly available benchmark datasets as listed in Table 2, ensuring that they are consistent with previous fine-tuning approaches. CNN/DailyMail (Hermann et al., 2015) is the most widely-adopted summarization dataset that contains news articles and corresponding highlights as summaries. We use the non-anonymized version and follow the common training/validation/testing splits (287,084/13,367/11,489). XSum (Narayan et al., 2018) is a one-sentence news summarization dataset with professionally written summaries. We follow the common splits (204,045/11,332/11,334). PubMed (Cohan et al., 2018) is a scientific paper summarization dataset and we use the introduction section as the article and the abstract section as the summary following (Zhong et al., 2020) with common splits (83,233/4,946/5,025). Reddit (Kim et al., 2018) is a highly abstractive dataset collected from social media platforms with a split (41,675/645/645).

**Evaluation:** We conducted an evaluation of ChatGPT's summarization performance utilizing ROUGE (Lin and Hovy, 2003) following previous studies. We also employ a GPT-based evaluation metric G-EVAL (Liu et al., 2023). To investigate the faithfulness of the summaries, we employed common metrics FactCC (Kryściński et al., 2019) and QuestEval (Scialom et al., 2021).

We selected the best prompts on a dev set of 50 examples and randomly sampled 1000 examples from each test set of the original dataset for evaluation. The detailed prompts used in the experiments and more details about the experimental setup can be found in Table 4 and Appendix B.

### 4.2 Experiments Results

The overall results are shown in Table 1. The upper block includes extractive results and SOTA scores from MatchSum (Zhong et al., 2020). The lower block includes abstractive results and SOTA scores from BRIO (Liu et al., 2022) for CNN/DM and XSum, SummaReranker (Ravaut et al., 2022) for

| Dataset | Setting | RL | G-EVAL | FactCC | QuestEval |
|---------|---------|------|--------|--------|-----------|
| Reddit | Abs | 18.54 | 2.43 | 9.46 | 40.79 |
| | Ext-Abs | 18.26 | 2.60 | **60.40** | **49.45** |
| | Oracle-Abs | **19.37** | **2.64** | 59.75 | 48.93 |
| XSum | Abs | 20.21 | 2.67 | 5.42 | 46.14 |
| | Ext-Abs | 18.55 | 2.28 | **55.73** | **53.25** |
| | Oracle-Abs | **21.10** | **2.72** | 55.03 | 53.21 |
| PubMed | Abs | 28.46 | 2.70 | 8.37 | 42.83 |
| | Ext-Abs | 26.50 | 2.81 | 26.38 | 44.32 |
| | Oracle-Abs | **26.51** | **2.83** | **27.35** | **44.50** |
| CNN/DM | Abs | 28.39 | 3.24 | 6.35 | 45.32 |
| | Ext-Abs | 29.16 | 3.50 | 51.65 | 51.72 |
| | Oracle-Abs | **33.32** | **3.51** | **53.67** | **52.46** |

Table 3: Summarization results of the extract-then-generate pipeline. Abs, Ext-Abs, and Oracle-Abs refer to the generate-only baseline, the extract-then-generate pipeline, and generation based on ORACLE, respectively.

Reddit, and GSum (Dou et al., 2020) for PubMed.

It is observed that ChatGPT generally achieves *lower* ROUGE scores in comparison to previous fine-tuning methods for all datasets under both extractive and abstractive settings, but achieves *higher* scores in terms of LLM-based evaluation metric G-EVAL. The findings are consistent with the previous conclusion in (Goyal et al., 2022; Zhang et al., 2023d). We also observe that ChatGPT-Ext outperforms ChatGPT-Abs in two extractive datasets CNN/DM and PubMed while performing worse in the other two abstractive datasets. We argue the results are due to the bias within the reference summaries of the dataset and the limit of ROUGE scores. Nonetheless, we notice that despite being primarily designed for generation tasks, ChatGPT achieves impressive results in extractive summarization, which requires comprehension of the documents. The decoder-only structure of ChatGPT doesn't degrade its comprehension capability compared to encoder models like BERT. We also find that the ROUGE score gap between ChatGPT and SOTA fine-tuned baselines are smaller in the extractive setting than in the abstractive setting.

The results also indicate that in-context learning and reasoning are generally beneficial for the extractive summarization task across four datasets in different domains. We only observe performance degradation for in-context learning on the XSum dataset. We argue that the degradation comes from the short ORACLE of XSum, which brings more confusion with a few ORACLE examples. However, with chain-of-thought reasoning explanations, ChatGPT can better understand the pattern and thus shows improvements with in-context reasoning. More in-context learning results could be found

in Table 5 in Appendix.

## 4.3 Extract Then Generate

We conduct further experiments to examine the effectiveness of the extract-then-generate framework as presented in Table 3.

The results show large improvements in summary factual consistency across all four datasets with the extract-then-generate framework. Notably, the FactCC scores are extremely low for generate-only baselines (less than 10 percent), highlighting the hallucination problems of ChatGPT-based summarization, where ChatGPT tends to make up new content in the summary. Nevertheless, the extract-then-generate framework effectively alleviates the hallucination problem of abstractive summaries by guiding the summary generation process with extracted salient sentences from the documents. We also find that guiding ChatGPT summary generation with its own extracted summaries leads to similar summary faithfulness improvements compared to guiding generation with ORACLE.

In terms of summary quality, the results demonstrate that the performance of ChatGPT improves largely in terms of ROUGE scores when grounded with the ORACLE summaries. However, the ROUGE score performance of the extract-then-generate framework relies heavily on the extractive performance when grounded with its own extractive summaries. In summary, the extract-then-generate framework could effectively improve the summary faithfulness with similar or even better summary quality.

### 4.4 Positional Bias

Lead bias is a common phenomenon in extractive summarization, especially in the news domain, where early parts of an article often contain the most salient information. As shown in Figure 1, we find that the position distribution of the ChatGPT extracted summary sentences is skewed towards a higher position bias than the ORACLE sentences. In addition, in-context learning brings more positional bias to the summaries. The results indicate that LLMs may rely on superficial features like sentence positions for extractive summarization.

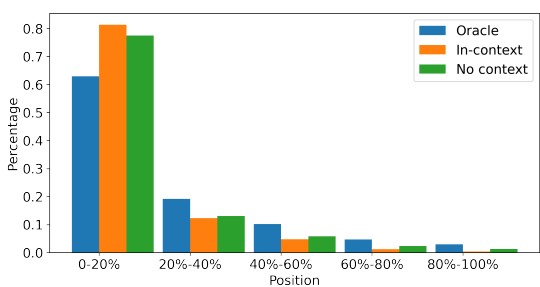

Figure 1: Position distribution of extracted sentences on 1000 random samples from the CNN/DM test set.

## 5 Conclusion

This paper presents a thorough evaluation of Chat-GPT's performance on extractive summarization across four benchmark datasets. The results indicate ChatGPT's strong potential for the task and the possibility of generating factual summaries using the extract-generate framework. Overall, this study suggests that ChatGPT is a powerful tool for text summarization, and we hope the insights gained from this work can guide future research in this area.

## Limitations

Instead of conducting experiments on the entire test set, we randomly sample 1000 examples from each dataset test set due to budget limits. Previous research efforts (Goyal et al., 2022; Zhang et al., 2023d) have also been limited in their testing of GPT-3 on a small number of instances.

Our experimental results are mainly evaluated with various automatic metrics (summary quality and faithfulness). We plan to include a human study to further verify the conclusions in the future.

We only use *gpt-3.5-turbo* model from openAI API as an instance of large language models. The emphasis of the paper is to explore extractive summarization and extract-then-generate pipeline with LLM, but not compare different open and closed LLMs.

## Acknowledgement

This work is partially supported by NSF through grants IIS-1763365 and IIS-2106972.

We express our gratitude to the anonymous reviewers for their valuable reviews and feedback.

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

## Appendix

## A Prompts

Here we list prompts used in our experiments for extracted and generated summaries in Table 4. Note that according to OpenAI's document, the model could receive two categories of prompts: system prompt and user prompt, where the system prompt functions as the global instruction to initialize the model and the user prompt as the question proposed by users.

## B Experimental Setup

We employed the gpt-3.5-turbo model[2] for the generation and assessment of summaries, maintaining a temperature setting of 0 to ensure reproducibility.

Regarding the datasets, a random sampling method was adopted where 1000 samples were chosen for each dataset for experimental purposes. Furthermore, a smaller subset of 50 samples was utilized for the discovery of optimal prompts and hyperparameters. The random seed was established at 101 to promote consistency.

In accordance with established research, the ROUGE[3] F-1 scores were implemented as the automatic evaluation metrics (Lin and Hovy, 2003). To be specific, the ROUGE-1/2 scores serve as measures of summary informativeness, while the ROUGE-L score gauges the fluency of the summary. In addition to these measures, a GPT model was integrated as a summary evaluator to mimic human evaluation processes. This evaluator was designed to assess the summary based on a comprehensive analysis of coherence, consistency, fluency, and efficiency. The findings from each experiment are reported as single-run results.

The experiments involving each dataset, which includes 1000 examples, will run for 1.5 hours to perform both inference and evaluation.

## C In-context Learning Results

The detailed in-context learning results are shown in Table 5

## D Document length Analysis

We further investigate the influence of document length on the summarization performance, as pre-

---

[2]https://platform.openai.com/docs/guides/gpt-completions-api

[3]ROUGE: https://pypi.org/project/rouge-score/

| Setting | Prompt |
|---|---|
| Extractive | **System**: You are an extractive summarizer that follows the output pattern. **User**: Please extract sentences as the summary. The summary should contain **m** sentences. Document: [*Test Document*] [*Format Instruction*]. |
| Abstractive | **System**: You are an abstractive summarize that follows the output pattern. **User**: Please write a summary for the document. Document: [*Test Document*] [*Format Instruction*] |
| In-context | **System**: You are an extractive summarizer that follows the output pattern. **User**: The following examples are successful extractive summarization instances: [*n Document-Summary Pairs*]. Please summarize the following document. Document: [*Test Document*]. The summary should contain m sentences. [*Format Instruction*]. |
| Explanation | **System**: You are an extractive summarizer that follows the output pattern. **User**: The following examples are successful extractive summarization instances: [*n Document-Summary-Reason Triads*]. Please summarize the following document and give the reason. Document: [*Test Document*]. The summary should contain m sentences. [*Format Instruction*]. |
| Extract-abstract | **System**: You are an abstractive summarizer that follows the output pattern. **User**: Please revise the extracted summary based on the document. The revised summary should include the information in the extracted summary. Document: [*Test Docuemnt*] Extractive Summary: [*Extractive Summary*] [*Format Instruction*]. |
| Evaluator | **System**: You are a summary evaluator that follows the output pattern. You give scores for the summaries based on the comprehensive consideration following criteria: (1) Coherence: "the collective quality of all sentences"; (2) Consistency: "the factual alignment between the summary and the reference"; (3) Fluency: " the quality of individual sentences"; (4) Efficiency: "If the summary is concise" **User**: Please evaluate the summary based on the reference summary.Reference:[*Reference Summary*] Summary:[*Predicted Summary*][*Format Instruction*]. |

Table 4: Prompts used for both extractive and abstractive summarization. $m$ is the number of extracted sentences defined in Table 2. Document-summary pairs and document-summary-reason triads are the input contexts. $n$ is the number of context instances.

| # Context | CNN/DM | | | XSum | | |
|---|---|---|---|---|---|---|
| | R1 | R2 | RL | R1 | R2 | RL |
| 0 | $39.25 \pm 0.23$ | $15.36 \pm 1.10$ | $25.90 \pm 0.97$ | $19.85 \pm 2.59$ | $2.96 \pm 2.59$ | $13.29 \pm 1.30$ |
| 1 | $40.62 \pm 0.70$ | $17.00 \pm 1.06$ | $26.44 \pm 0.84$ | $15.33 \pm 0.50$ | $2.48 \pm 0.19$ | $11.48 \pm 0.13$ |
| 1w/R | $38.83 \pm 0.91$ | $14.94 \pm 2.53$ | $25.36 \pm 1.82$ | $17.86 \pm 1.73$ | $3.29 \pm 0.85$ | $12.55 \pm 1.29$ |
| 2 | $40.91 \pm 0.69$ | $15.68 \pm 0.61$ | $26.13 \pm 0.83$ | $18.61 \pm 0.39$ | $4.42 \pm 0.97$ | $14.06 \pm 2.01$ |
| 2w/R | $41.70 \pm 0.70$ | $15.95 \pm 0.92$ | $26.98 \pm 1.33$ | $17.95 \pm 3.03$ | $4.11 \pm 1.01$ | $13.46 \pm 1.76$ |
| 3 | $\mathbf{42.38 \pm 0.13}$ | $17.27 \pm 0.23$ | $\mathbf{28.41 \pm 0.31}$ | $17.49 \pm 1.87$ | $3.86 \pm 1.55$ | $12.94 \pm 2.16$ |
| 3w/R | $42.26 \pm 1.38$ | $17.02 \pm 1.60$ | $27.42 \pm 1.62$ | $\mathbf{20.37 \pm 1.61}$ | $\mathbf{4.78 \pm 0.44}$ | $\mathbf{14.21 \pm 1.07}$ |
| 4 | $42.26 \pm 0.50$ | $\mathbf{17.41 \pm 0.83}$ | $27.96 \pm 0.83$ | $16.68 \pm 1.56$ | $3.72 \pm 0.20$ | $12.12 \pm 1.19$ |
| 4w/R | $41.23 \pm 0.93$ | $17.08 \pm 0.38$ | $28.25 \pm 0.93$ | $18.17 \pm 0.28$ | $4.05 \pm 0.38$ | $12.74 \pm 0.94$ |
| 5 | $40.71 \pm 1.92$ | $16.96 \pm 0.91$ | $27.42 \pm 1.26$ | $17.43 \pm 1.08$ | $3.53 \pm 0.96$ | $12.33 \pm 0.51$ |
| 5w/R | $40.18 \pm 0.83$ | $15.15 \pm 1.44$ | $25.98 \pm 1.91$ | $19.55 \pm 0.64$ | $4.29 \pm 0.46$ | $13.13 \pm 0.68$ |

Table 5: In-context learning experimental results on CNN/DM and XSum datasets. For each dataset, we randomly sampled 50 data from the test set. In each section, w/R means we provide human written reasons for each context document. For the test document, we also ask the system to generate the reason why it choose selected sentences.

sented in Figure 2. Our findings suggest that Chat-GPT maintains consistent performance across documents of different lengths, indicating the model's robustness in the context of extractive summarization.

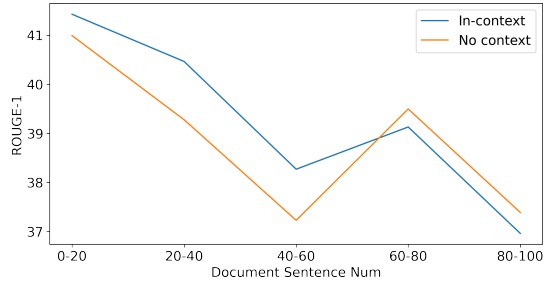

Figure 2: ROUGE-1 Score distribution over document of different lengths.

# E   Case Study

Here we show the ChatGPT-generated summaries with different prompt settings in Table 6 for one example from the CNNDM dataset.

| | |
|---|---|
| **Document** | Daredevil Nik Wallenda says he'll walk untethered on top of a 400-foot observation wheel in Orlando, Florida, this month. Wallenda said Monday at a New York City news conference that the Orlando Eye will be moving when he attempts his feat on April 29. The Orlando Eye, part of a new entertainment complex, will offer views of central Florida from inside 30 enclosed, air-conditioned glass capsules when it opens to the public on May 4. Eyes on the prize: high-wire performer Nik Wallenda announces his latest stunt at the 400-foot Orlando Eye, during a news conference, in New York on Monday. Tough challenge: the 36-year-old daredevil will walk atop the Orlando Eye as it turns on April 29. The Orlando Eye team issued a statement saying it's excited to have Wallenda attempt the 'amazing stunt.' No distance for the performance has been set yet, but Wallenda, 36, said he was not likely to walk the entire 20 minutes or so that it takes the wheel to go a full circle. Wallenda previously walked atop a Ferris wheel in Santa Cruz, California, but he said the size of the much larger Orlando wheel and the fact that he will not use a pole sets this stunt apart. The seventh-generation member of the 'Flying Wallenda' family of acrobats has walked across the Grand Canyon and Niagara Falls. In November, he walked twice between two Chicago skyscrapers without a net or harness, doing the second walk blindfolded. Wallenda is the great-grandson of Karl Wallenda, who fell to his death in Puerto Rico at 73. |
| **Reference** | The 36-year-old will stage his next stunt on April 29. In November, Wallenda walked back and forth between two Chicago skyscrapers in a live television event. His great-grandfather Karl Wallenda died in a tightrope walk in Puerto Rico in 1978. Wallenda has also tightrope walked across Niagara Falls and the Grand Canyon. |
| **ORACLE** | Tough challenge: the 36-year-old daredevil will walk atop the Orlando Eye as it turns on April 29. The seventh-generation member of the 'Flying Wallenda' family of acrobats has walked across the Grand Canyon and Niagara Falls. In November, he walked twice between two Chicago skyscrapers without a net or harness, doing the second walk blindfolded. |
| **Abstractive** | Daredevil Nik Wallenda plans to walk untethered atop the 400-foot Orlando Eye observation wheel on April 29th. The 36-year-old high-wire performer, a seventh-generation member of the "Flying Wallenda" acrobat family, has previously completed daring walks across the Grand Canyon, Niagara Falls, and between Chicago skyscrapers. Although the exact distance for this stunt hasn't been determined, Wallenda stated he's unlikely to walk the entire 20 minutes it takes for the wheel to complete a full circle. |
| **Extractive** | Daredevil Nik Wallenda says he'll walk untethered on top of a 400-foot observation wheel in Orlando, Florida, this month. Wallenda said Monday at a New York City news conference that the Orlando Eye will be moving when he attempts his feat on April 29. The 36-year-old daredevil will walk atop the Orlando Eye as it turns on April 29. |
| **In-Context** | Daredevil Nik Wallenda says he'll walk untethered on top of a 400-foot observation wheel in Orlando, Florida, this month. Wallenda said Monday at a New York City news conference that the Orlando Eye will be moving when he attempts his feat on April 29. Wallenda is the great-grandson of Karl Wallenda, who fell to his death in Puerto Rico at 73. |
| **Extract-abstract** | Nik Wallenda, a daredevil and seventh-generation member of the "Flying Wallenda" family of acrobats, will attempt to walk untethered on top of a 400-foot observation wheel in Orlando, Florida, on April 29. The Orlando Eye, part of a new entertainment complex, will offer views of central Florida from inside 30 enclosed, air-conditioned glass capsules when it opens to the public on May 4. Wallenda previously walked atop a Ferris wheel in Santa Cruz, California, but he said the size of the much larger Orlando wheel and the fact that he will not use a pole sets this stunt apart. Wallenda is the great-grandson of Karl Wallenda, who fell to his death in Puerto Rico at 73. |

Table 6: Case study of different settings