# OpenReview forum: "Extractive Summarization via ChatGPT for Faithful Summary Generation"
_EMNLP/2023/Conference — EMNLP 2023 Findings_

### Official Review · Reviewer_hwbk · 2023-08-02

**Soundness:** 3

**Excitement:**

4: Strong: This paper deepens the understanding of some phenomenon or lowers the barriers to an existing research direction.

**Paper Topic And Main Contributions:**

This study explores the potential of ChatGPT for text summarization, specifically in an extractive setting. The findings suggest that while ChatGPT achieves comparably lower ROUGE scores (compared to fine-tuning methods), it performs better in LM-based automatic evaluation scores. In the follow-up discussion, the authors found that ChatGPT exhibits higher performance improvement in terms of faithfulness when prompted to perform an extract-then-generate procedure for generating summaries. The experiments on four different and diverse datasets indicate the high potential of ChatGPT in text summarization, as the authors suggest.

**Reasons To Accept:**

- The paper touches on an interesting and significant topic in the context of text summarization, which could make a substantial contribution to the field.
- The findings, specifically regarding the potential use of ChatGPT in producing faithful summaries when prompted with an extract-then-generate approach, are particularly interesting.
- The experiments have been conducted on a handful of datasets from diverse domains (general news, scientific, and social media), indicating the strengths of the proposed arguments in different domains.

**Reasons To Reject:**

- The paper must be re-structured. The authors have introduced some concepts such as "in-context" and "chain-of-thought" learning/reasoning in the introduction for the first time but haven't provided any explanation of what these are. While an attempt has been made to explain "in-context learning" in section 3.2, the reader still gets confused about the "chain-of-thought" approach as it has never been explained in the paper. This adds to the lack of clarity of the paper, as a significant part of the results are based on these learning approaches as they stand.

- While the authors have mentioned that ChatGPT underperforms the fine-tuning methods on the ROUGE metric and outperforms them on LM-based metrics, they have not provided the underlying reasons for this inconsistency.

- The paper could make a more substantial contribution with a human analysis on a subset of the results, as automatic metrics may not be sufficient.

- The reader would like to see a discussion of choosing ChatGPT over the fine-tuning methods. Although this may involve a trade-off, having a discussion on this topic would add to the contributions of the paper.

**Reproducibility:**

2: Would be hard pressed to reproduce the results. The contribution depends on data that are simply not available outside the author's institution or consortium; not enough details are provided.

**Reviewer Confidence:**

4: Quite sure. I tried to check the important points carefully. It's unlikely, though conceivable, that I missed something that should affect my ratings.

**Typos Grammar Style And Presentation Improvements:**

- Task formulation section is not needed as we all know what extractive summarization is, let alone the paper is short and has limited space to share the findings. I hence suggest removing it to have more space for other important sections such as discussion or introducing "in-context" and "chain-of-thought" concepts.

---

> ### Author Rebuttal · Authors · 2023-08-29
>
> Thanks for your thorough review of our paper and the effort you've put into providing valuable feedback. We have carefully considered your concern, and we would like to address comprehensively:
>
> **Question: Task formulation section is not needed as we all know what extractive summarization is, let alone the paper is short and has limited space to share the findings. I hence suggest removing it to have more space for other important sections such as discussion or introducing "in-context" and "chain-of-thought" concepts.**
>
> **Answer:** Thanks for pointing this out. Chain-of-thought is a recent prompting technique to improve the reasoning capabilities of LLM by generating a chain of thought -- a series of intermediate reasoning steps. It is widely used in prompting LLM and we will definitely add more information in the final version of the paper.
>
>
> **Question: While the authors have mentioned that ChatGPT underperforms the fine-tuning methods on the ROUGE metric and outperforms them on LM-based metrics, they have not provided the underlying reasons for this inconsistency.**
>
> **Answer:** The major reason for this inconsistency is the fine-tuned models learn data-specific patterns during the fine-tuning process. In contrast, ChatGPT is under zero/few-shot setting. We will clarify this in the final draft of the paper.
>
> **Question: The paper could make a more substantial contribution with a human analysis on a subset of the results, as automatic metrics may not be sufficient.**
>
>
> **Answer:** We agree that human analysis will better demonstrate the performance of different systems. The human annotation results are summarized in the following table on both CNNDM and XSUM datasets under the zero-shot settings. The aspect-based ratings are from a scale of 1 to 5, and the human preference is the percentage of the summary being the best system.
>
>
> **Human Eval for CNN/DM datasets**
>
> | Model | Coherence | Fluency | Relevance | Consistency | Conciseness | Overall | Human Pref |
> |--------------|--------------------|------------------|--------------------|----------------------|----------------------|------------------|------------------------------------|
> | BART         | 3.92               | 4.16             | 4.00               | 3.12                 | 3.64                 | 3.24             | 0.04                        |
> | T5           | 3.72               | 4.24             | 4.32      | 3.52                 | 3.84                 | 3.68             | 0.10                               |
> | PEGASUS      | 3.20               | 3.53             | 3.33               | 2.87                 | 1.85                 | 1.63             | 0.00                               |
> | ChatGPT      | 4.20               | 4.36             | 4.28               | 4.01                 | 3.92        | 4.01             | 0.34                               |
>
>
>
> **Human Eval for XSum datasets**
>
> | Model | Coherence | Fluency | Relevance | Consistency | Conciseness | Overall | Human Pref |
> |--------------|--------------------|------------------|--------------------|----------------------|----------------------|------------------|------------------------------------|
> | BART         | 3.97               | 4.30             | 4.13               | 3.30                 | 3.93        | 3.84             | 0.30                       |
> | T5           | 3.84               | 4.32             | 4.02               | 3.63                 | 3.84                 | 3.25             | 0.08                               |
> | PEGASUS      | 3.13               | 4.10             | 3.52               | 2.87                 | 2.03                 | 2.41             | 0.00                               |
> | ChatGPT      | 4.03               | 4.40    | 4.30      | 3.93                 | 3.87                 | 3.92             | 0.24                               |
>
> According to the results, human evaluation generally aligns with the GPT evaluation and ChatGPT outperforms other models by a large margin.
>
>  We used Pearson Correlation to evaluate the inter-annotator agreement:
>
> |                 |       |
> | --------------- | ----- |
> | **Coherence**   | 0.658 |
> | **Fluency**     | 0.851 |
> | **Relevance**   | 0.796 |
> | **Consistency** | 0.621 |
> | **Conciseness** | 0.747 |
> | **Overall**     | 0.742 |
>
> We will add all these human studies in our final version.
>
> **Question: The reader would like to see a discussion of choosing ChatGPT over the fine-tuning methods. Although this may involve a trade-off, having a discussion on this topic would add to the contributions of the paper.**
>
> **Answer:** As mentioned above, the major limitation of ChatGPT is that we could not fine-tune it with large-scale in-domain data. We experimented with LLaMa [1], one open LLM model on the Reddit dataset, and got ROUGE numbers of 26.24/6.82/21.58. We will add more results in our final draft.
>
> [1]LLaMA: Open and Efficient Foundation Language Models

---

### Official Review · Reviewer_jH6Y · 2023-08-04

**Soundness:** 3

**Excitement:**

3: Ambivalent: It has merits (e.g., it reports state-of-the-art results, the idea is nice), but there are key weaknesses (e.g., it describes incremental work), and it can significantly benefit from another round of revision. However, I won't object to accepting it if my co-reviewers champion it.

**Paper Topic And Main Contributions:**

This paper provides a preliminary experimental test of the effectiveness of ChatGPT on the task of extractive summarization and verifies that the factual consistency of ChatGPT-generated summaries can be further improved by adopting the extract-generate framework.

**Reasons To Accept:**

The paper is generally well-written and the evaluation of ChatGPT’s performance on extractive summarization is interesting and well explained.

**Reasons To Reject:**

1. Evaluating the effectiveness of large models on extractive summarization tasks using only ChatGPT as a representative seems insufficient.
2. It is not meaningful to compare the factual consistency of the generated and extracted summaries in the experiments (e.g., Table 3), because all the content of the extracted summaries comes from the original text, so it is obvious that the factual consistency is better, and the use of the joint extraction and generation model can significantly improve the quality of summary generation, which is one of the hot topics in recent years, and this paper's work on the "extract-generate framework" is not significantly innovative.
3. Given the complexity of summary quality assessment, it is still necessary and recommended to conduct the manual evaluation.

**Reproducibility:**

4: Could mostly reproduce the results, but there may be some variation because of sample variance or minor variations in their interpretation of the protocol or method.

**Reviewer Confidence:**

3: Pretty sure, but there's a chance I missed something. Although I have a good feel for this area in general, I did not carefully check the paper's details, e.g., the math, experimental design, or novelty.

---

> ### Author Rebuttal · Authors · 2023-08-29
>
> Thanks for your thorough review of our paper and the effort you've put into providing valuable feedback. We have carefully considered your concern, and we would like to address comprehensively:
>
> **Question: Given the complexity of summary quality assessment, it is still necessary and recommended to conduct the manual evaluation.**
>
> **Answer:** We agree that human analysis will better demonstrate the performance of different systems. The human annotation results are summarized in the following table on both CNNDM and XSUM datasets under the zero-shot settings. The aspect-based ratings are from a scale of 1 to 5, and the human preference is the percentage of the summary being the best system.
>
>
> **Human Eval for CNN/DM datasets**
>
> | Model | Coherence | Fluency | Relevance | Consistency | Conciseness | Overall | Human Pref |
> |--------------|--------------------|------------------|--------------------|----------------------|----------------------|------------------|------------------------------------|
> | BART         | 3.92               | 4.16             | 4.00               | 3.12                 | 3.64                 | 3.24             | 0.04                        |
> | T5           | 3.72               | 4.24             | 4.32      | 3.52                 | 3.84                 | 3.68             | 0.10                               |
> | PEGASUS      | 3.20               | 3.53             | 3.33               | 2.87                 | 1.85                 | 1.63             | 0.00                               |
> | ChatGPT      | 4.20               | 4.36             | 4.28               | 4.01                 | 3.92        | 4.01             | 0.34                               |
>
>
>
> **Human Eval for XSum datasets**
>
> | Model | Coherence | Fluency | Relevance | Consistency | Conciseness | Overall | Human Pref |
> |--------------|--------------------|------------------|--------------------|----------------------|----------------------|------------------|------------------------------------|
> | BART         | 3.97               | 4.30             | 4.13               | 3.30                 | 3.93        | 3.84             | 0.30                       |
> | T5           | 3.84               | 4.32             | 4.02               | 3.63                 | 3.84                 | 3.25             | 0.08                               |
> | PEGASUS      | 3.13               | 4.10             | 3.52               | 2.87                 | 2.03                 | 2.41             | 0.00                               |
> | ChatGPT      | 4.03               | 4.40    | 4.30      | 3.93                 | 3.87                 | 3.92             | 0.24                               |
>
> According to the results, human evaluation generally aligns with the GPT evaluation and ChatGPT outperforms other models by a large margin.
>
>  We used Pearson Correlation to evaluate the inter-annotator agreement:
>
> |                 |       |
> | --------------- | ----- |
> | **Coherence**   | 0.658 |
> | **Fluency**     | 0.851 |
> | **Relevance**   | 0.796 |
> | **Consistency** | 0.621 |
> | **Conciseness** | 0.747 |
> | **Overall**     | 0.742 |
>
> We will add all these human studies in our final version.
>
>
> **Question: Evaluating the effectiveness of large models on extractive summarization tasks using only ChatGPT as a representative seems insufficient.**
>
> **Answer:** Thanks for pointing this out. By the time we prepared this draft for submission, ChatGPT is still one of the strongest large language models. We will add more recent LLM like GPT-4 and LLaMa in our final version.

---

### Official Review · Reviewer_6bTC · 2023-08-12

**Soundness:** 2

**Excitement:**

2: Mediocre: This paper makes marginal contributions (vs non-contemporaneous work), so I would rather not see it in the conference.

**Missing References:**

1. Bottom-Up Abstractive Summarization

2. Guiding Large Language Models via Directional Stimulus Prompting

**Paper Topic And Main Contributions:**

This paper presents an evaluation of ChatGPT's performance on extractive summarization on benchmark datasets. The paper also explores the effectiveness of in-context learning and chain-of-thought for enhancing the performance of ChatGPT in extractive summarization tasks. The results show that ChatGPT exhibits inferior extractive summarization performance in terms of ROUGE scores, but achieves higher performance on advance metrics, like LLM evaluation (G-EVAL). The authors also show that using an extract-then-generate pipeline with ChatGPT improves summary faithfulness compared to abstractive baselines.

**Questions For The Authors:**

A.	In table 2, how the number of sentences to extract is computed？

B.	When prompt ChatGPT to extract sentences, does all the output are original sentences from the document without any change?

C.	Why CoT doesn't work for summarization (Table 1) and sometimes hurt the performance?


**Reasons To Accept:**

1.	The paper focus on extractive summarization analyses with ChatGPT, which hasn't been discussed much yet. Extractive summarization is an important research direction that will contribute to factuality and simplification and has many application scenarios in real life.
2.	The observation that using an extract-then-generate pipeline with ChatGPT improves is consistent with intuition, providing practical insights for few-shot summarization.


**Reasons To Reject:**

1.	This analytical paper makes some contributions but lacks novelty. Combining extractive and abstractive methods has been proven effective in previous work. This has also been explored based on LLM.
2.	The experiment results are not very strong. The paper claims the advance and potential extract-then-generate pipeline with ChatGPT. But in Table 3, (1) the scores show limited improvement compared to Table 1; (2) the factual metrics only compare to abstractive methods but not extractive methods.  This method requires more in-depth analysis.
3.	Table 4 shows the prompts used for both extractive and abstractive summarization. (1) The extract-then-generate (Line 6) pipeline uses an extractive summary, which is generated by sentence-level extraction (Line 1). This can be a very coarse granularity and hurts flexibility.
(2) The prompts ask ChatGPT to ‘extract sentences’ or ‘write a summary’ to implement extractive/abstractive summarization. But actually, ChatGPT directly generates all summaries token by token. This method may not meet the strict, classic definition of extractive summarization.
(3) The evaluator is the same ChatGPT. There may be bias in evaluating the performance of models with the same model. This paper gives no more explanation.


**Reproducibility:**

4: Could mostly reproduce the results, but there may be some variation because of sample variance or minor variations in their interpretation of the protocol or method.

**Reviewer Confidence:**

3: Pretty sure, but there's a chance I missed something. Although I have a good feel for this area in general, I did not carefully check the paper's details, e.g., the math, experimental design, or novelty.

---

> ### Author Rebuttal · Authors · 2023-08-29
>
> Thanks for your thorough review of our paper and the effort you've put into providing valuable feedback. We have carefully considered your concern, and we would like to address comprehensively:
>
> **Question: In table 2, how the number of sentences to extract is computed？**
>
> **Answer:** The number are calculated based on the average number of sentences in the reference summaries in the development set. We use the same number with previous extractive summarization baseline papers like [1][2]. We will add these details in our final version.
>
>
> **Question: The experiment results are not very strong. The paper claims the advance and potential extract-then-generate pipeline with ChatGPT. But in Table 3, (1) the scores show limited improvement compared to Table 1; (2) the factual metrics only compare to abstractive methods but not extractive methods. This method requires more in-depth analysis.**
>
> **Question: Table 4 shows the prompts used for both extractive and abstractive summarization. (1) The extract-then-generate (Line 6) pipeline uses an extractive summary, which is generated by sentence-level extraction (Line 1). This can be a very coarse granularity and hurts flexibility. (2) The prompts ask ChatGPT to ‘extract sentences’ or ‘write a summary’ to implement extractive/abstractive summarization. But actually, ChatGPT directly generates all summaries token by token. This method may not meet the strict, classic definition of extractive summarization. (3) The evaluator is the same ChatGPT. There may be bias in evaluating the performance of models with the same model. This paper gives no more explanation.**
>
>
> **Answer:** ChatGPT has a strong instruction following capabilities. In the generation prompt, we use "Please extract sentences as the
> summary. The summary should contain
> m sentences. Document: [Test Document] [Format Instruction]. " and surprsingly found CHATGPT could extract sentences accurately without error during the generation, so it doesn't violate the strict, classic definition of extractive summarization.
>
> We agree that ChatGPT as evaluator may cause some self-bias, so we also conducted a **human study.** The human annotation results are summarized in the following table on both CNNDM and XSUM datasets under the zero-shot settings. The aspect-based ratings are from a scale of 1 to 5, and the human preference is the percentage of the summary being the best system.
>
>
> **Human Eval for CNN/DM datasets**
>
> | Model | Coherence | Fluency | Relevance | Consistency | Conciseness | Overall | Human Pref |
> |--------------|--------------------|------------------|--------------------|----------------------|----------------------|------------------|------------------------------------|
> | BART         | 3.92               | 4.16             | 4.00               | 3.12                 | 3.64                 | 3.24             | 0.04                        |
> | T5           | 3.72               | 4.24             | 4.32      | 3.52                 | 3.84                 | 3.68             | 0.10                               |
> | PEGASUS      | 3.20               | 3.53             | 3.33               | 2.87                 | 1.85                 | 1.63             | 0.00                               |
> | ChatGPT      | 4.20               | 4.36             | 4.28               | 4.01                 | 3.92        | 4.01             | 0.34                               |
>
>
>
> **Human Eval for XSum datasets**
>
> | Model | Coherence | Fluency | Relevance | Consistency | Conciseness | Overall | Human Pref |
> |--------------|--------------------|------------------|--------------------|----------------------|----------------------|------------------|------------------------------------|
> | BART         | 3.97               | 4.30             | 4.13               | 3.30                 | 3.93        | 3.84             | 0.30                       |
> | T5           | 3.84               | 4.32             | 4.02               | 3.63                 | 3.84                 | 3.25             | 0.08                               |
> | PEGASUS      | 3.13               | 4.10             | 3.52               | 2.87                 | 2.03                 | 2.41             | 0.00                               |
> | ChatGPT      | 4.03               | 4.40    | 4.30      | 3.93                 | 3.87                 | 3.92             | 0.24                               |
>
> According to the results, human evaluation generally aligns with the GPT evaluation and ChatGPT outperforms other models by a large margin.
>
> We used Pearson Correlation to evaluate the inter-annotator agreement:
>
> |                 |       |
> | --------------- | ----- |
> | **Coherence**   | 0.658 |
> | **Fluency**     | 0.851 |
> | **Relevance**   | 0.796 |
> | **Consistency** | 0.621 |
> | **Conciseness** | 0.747 |
> | **Overall**     | 0.742 |
>
> We will add all these human studies in our final version.
>
>
> **Question: When prompt ChatGPT to extract sentences, does all the output are original sentences from the document without any change?**
>
> **Answer:** As mentioned above, ChatGPT has a strong instruction following capabilities. Surprsingly we found that CHATGPT could extract sentences accurately without error during the generation.
>
>
> **Why CoT doesn't work for summarization (Table 1) and sometimes hurt the performance?**
>
> **Answer:** This is a fairly hard question to answer given ChatGPT is a black-box model. We noticed CoT brings very small degragation on CNNDM and larger on Reddit compared to in-context learning. One possible reason is our human reasoning reference is not ideal and brings more confusion to the model.
>
>
>
> [1]Text Summarization with Pretrained Encoders
>
> [2]Extractive Summarization as Text Matching

---

### Official Review · Reviewer_dGKd · 2023-08-12

**Paper Topic And Main Contributions:** 1) This study represents the first at…
**Soundness:** 3

**Excitement:**

3: Ambivalent: It has merits (e.g., it reports state-of-the-art results, the idea is nice), but there are key weaknesses (e.g., it describes incremental work), and it can significantly benefit from another round of revision. However, I won't object to accepting it if my co-reviewers champion it.

**Reasons To Accept:**

1. The paper is well written and provide enough details like prompt for reproducibility.
2. The authors have conducted through experiments and analysis. For example, conduct experiments on 4 benchmark datasets and evaluated using different automatic metrics.
3. The insights gained in the paper can be helpful to further research. For example, extract-then-abstract can be used to improve factuality.

**Reasons To Reject:**

The paper lacking fair baseline methods to compare with. The authors say their results on evaluated on randomly selected 1000 results. However, some the SOTA baselines as Table 1. are based on the full dataset.
The paper did not do human examination to better validate the improvement in factuality or overall quality. Also based on GPT-based metrics, GPT4 will provide more accurate evaluation then ChatGPT.
ChatGPT is essentially a generative method, it may be not accurate enough in copying the original sentences from the articles, I did not see how the authors are measuring and addressing this problem.

**Reproducibility:**

5: Could easily reproduce the results.

**Reviewer Confidence:**

4: Quite sure. I tried to check the important points carefully. It's unlikely, though conceivable, that I missed something that should affect my ratings.

---

> ### Author Rebuttal · Authors · 2023-08-29
>
> We sincerely value the priceless evaluation of our paper and the devoted time you've dedicated to offering valuable feedback. We have thoroughly assessed each of your points and are eager to provide thorough responses that address them comprehensively:
>
> **Question:The paper lacking fair baseline methods to compare with. The authors say their results on evaluated on randomly selected 1000 results. However, some the SOTA baselines as Table 1. are based on the full dataset. The paper did not do human examination to better validate the improvement in factuality or overall quality. Also based on GPT-based metrics, GPT4 will provide more accurate evaluation then ChatGPT. ChatGPT is essentially a generative method, it may be not accurate enough in copying the original sentences from the articles, I did not see how the authors are measuring and addressing this problem.**
>
> **Answer:** Thanks for pointing this out. Unfortunately due to budget and expense limits, we couldn't test CHATGPT on the full test sets on the large scale datasets. We expected using 1000 sampel test exmaples would also give a good approximation of the summarization performacne of ChatGPT.
>
>
>  We agree that **human examination** will better demonstrate the performance of different systems. The human annotation results are summarized in the following table on both CNNDM and XSUM datasets under the zero-shot settings. The aspect-based ratings are from a scale of 1 to 5, and the human preference is the percentage of the summary being the best system.
>
>
> **Human Eval for CNN/DM datasets**
>
> | Model | Coherence | Fluency | Relevance | Consistency | Conciseness | Overall | Human Pref |
> |--------------|--------------------|------------------|--------------------|----------------------|----------------------|------------------|------------------------------------|
> | BART         | 3.92               | 4.16             | 4.00               | 3.12                 | 3.64                 | 3.24             | 0.04                        |
> | T5           | 3.72               | 4.24             | 4.32      | 3.52                 | 3.84                 | 3.68             | 0.10                               |
> | PEGASUS      | 3.20               | 3.53             | 3.33               | 2.87                 | 1.85                 | 1.63             | 0.00                               |
> | ChatGPT      | 4.20               | 4.36             | 4.28               | 4.01                 | 3.92        | 4.01             | 0.34                               |
>
>
>
> **Human Eval for XSum datasets**
>
> | Model | Coherence | Fluency | Relevance | Consistency | Conciseness | Overall | Human Pref |
> |--------------|--------------------|------------------|--------------------|----------------------|----------------------|------------------|------------------------------------|
> | BART         | 3.97               | 4.30             | 4.13               | 3.30                 | 3.93        | 3.84             | 0.30                       |
> | T5           | 3.84               | 4.32             | 4.02               | 3.63                 | 3.84                 | 3.25             | 0.08                               |
> | PEGASUS      | 3.13               | 4.10             | 3.52               | 2.87                 | 2.03                 | 2.41             | 0.00                               |
> | ChatGPT      | 4.03               | 4.40    | 4.30      | 3.93                 | 3.87                 | 3.92             | 0.24                               |
>
> According to the results, human evaluation generally aligns with the GPT evaluation and ChatGPT outperforms other models by a large margin.
>
>  We used Pearson Correlation to evaluate the inter-annotator agreement:
> |                 |       |
> | --------------- | ----- |
> | **Coherence**   | 0.658 |
> | **Fluency**     | 0.851 |
> | **Relevance**   | 0.796 |
> | **Consistency** | 0.621 |
> | **Conciseness** | 0.747 |
> | **Overall**     | 0.742 |
>
> We will add all these human studies in our final version.
>
> In terms of **extractive summarization of CHATGPT**, it has a strong instruction following capabilities. In the generation prompt, we use "Please extract sentences as the
> summary. The summary should contain
> m sentences. Document: [Test Document] [Format Instruction]. " and found CHATGPT could extract sentences accurately without error during the generation.

---

### Meta-Review · Area_Chair_iAgd · 2023-09-18

**Recommendation:** 3

**Metareview:**

The paper provides a thorough analysis of ChatGPT's performance on extractive summarization and contrasts it with conventional fine-tuning techniques. The study is performed on multiple benchmark datasets.  The experimental results show that ChatGPT performs better on LLM-based assessment measures than existing supervised systems, but performs worse on extractive summarization tasks as measured by ROUGE scores. The authors also explore the effectiveness of in-context learning and chain-of-thought reasoning for enhancing chatGPT's performance on extractive summarization. Finally, the research reveals that an extract-then-generate pipeline with ChatGPT significantly outperforms abstractive baselines in terms of summary faithfulness.

The following pros were revealed by reviewers:
1. It is the first attempt to extend the application of ChatGPT to extractive summarization and evaluate its performance.
2. Investigation of the effectiveness of in-context learning and chain-of-thought reasoning for extractive summarization using ChatGPT.
3. Experiments with the extract-then-generate framework that significantly outperforms the generated summary faithfulness compared to abstractive baselines.
4. The paper is well-written and the experiments with ChatGPT for extractive summarization are well organized and explained.

Some drawbacks were also noticed, such as using only one LLM and a limited number of documents in the experiments. The authors addressed all the comments and justified their choices.

---

### Decision · Program_Chairs · 2023-10-07

**Decision:**

Accept-Findings

**Comment:**

The paper provides a thorough analysis of ChatGPT's performance on extractive summarization and contrasts it with conventional fine-tuning techniques. The study is performed on multiple benchmark datasets.  The experimental results show that ChatGPT performs better on LLM-based assessment measures than existing supervised systems, but performs worse on extractive summarization tasks as measured by ROUGE scores. The authors also explore the effectiveness of in-context learning and chain-of-thought reasoning for enhancing chatGPT's performance on extractive summarization. Finally, the research reveals that an extract-then-generate pipeline with ChatGPT significantly outperforms abstractive baselines in terms of summary faithfulness.

The following pros were revealed by reviewers:
1. It is the first attempt to extend the application of ChatGPT to extractive summarization and evaluate its performance.
2. Investigation of the effectiveness of in-context learning and chain-of-thought reasoning for extractive summarization using ChatGPT.
3. Experiments with the extract-then-generate framework that significantly outperforms the generated summary faithfulness compared to abstractive baselines.
4. The paper is well-written and the experiments with ChatGPT for extractive summarization are well organized and explained.

Some drawbacks were also noticed, such as using only one LLM and a limited number of documents in the experiments. The authors addressed all the comments and justified their choices.